# Optimization of energy and time predicts dynamic speeds for human walking

Rebecca Elizabeth Carlisle[1], Arthur D Kuo[1,2]*

[1]Biomedical Engineering Program, University of Calgary, Calgary, Canada; [2]Faculty of Kinesiology, University of Calgary, Calgary, Canada

**Abstract** Humans make a number of choices when they walk, such as how fast and for how long. The preferred steady walking speed seems chosen to minimize energy expenditure per distance traveled. But the speed of actual walking bouts is not only steady, but rather a time-varying trajectory, which can also be modulated by task urgency or an individual's movement vigor. Here we show that speed trajectories and durations of human walking bouts are explained better by an objective to minimize Energy and Time, meaning the total work or energy to reach destination, plus a cost proportional to bout duration. Applied to a computational model of walking dynamics, this objective predicts dynamic speed vs. time trajectories with inverted U shapes. Model and human experiment (N=10) show that shorter bouts are unsteady and dominated by the time and effort of accelerating, and longer ones are steadier and faster and dominated by steady-state time and effort. Individual-dependent vigor may be characterized by the energy one is willing to spend to save a unit of time, which explains why some may walk faster than others, but everyone may have similar-shaped trajectories due to similar walking dynamics. Tradeoffs between energy and time costs can predict transient, steady, and vigor-related aspects of walking.

## Editor's evaluation

This valuable study presents a new optimal control cost framework to predict features of walking bouts, adding a cost function term proportional to the duration of the walking bout in addition to the conventional energetic expenditure term. The authors show solid evidence that predicted optimal trajectories from simulations qualitatively match walking data from human subjects. This study provides a foundation for future studies testing the validity of the energy-time hypothesis for human gait speed selection across different walking bouts in able-bodied and patient populations.

*For correspondence:
arthurdkuo@gmail.com

Competing interest: The authors declare that no competing interests exist.

## Introduction

Many aspects of human walking are determined by minimization of metabolic energy expenditure. For example, the preferred step length (*Atzler and Herbst, 1927*) and step width (*Donelan et al., 2001*) minimize energy expenditure for a given steady speed, and the preferred steady speed approximately coincides with minimum energy expenditure per distance traveled (*Figure 1A*, *Ralston, 1958*). This speed, as well as the economy of walking, both decline with age, disability, or poor health. As such, preferred speed is widely employed as a clinically useful indicator of overall mobility (*Afilalo et al., 2010*; *Studenski et al., 2011*). However, there are naturally many other factors that also influence walking. All walking tasks have a beginning and end, and some may spend little or no time at steady speed. Some tasks may also occur with a degree of urgency, and some individuals may habitually walk faster than others, for reasons not obviously explained by economy. Energy economy is a powerful and objective explanation for steady walking speed, but it does not readily accommodate

**Figure 1.** Humans prefer an economical speed for steady walking, but not all walking is steady. (**A**) The preferred steady walking speed $v^*$ coincides with minimum metabolic cost of transport ('min COT'), which has a convex dependence on speed (after *Ralston, 1958*). (**B**) The distribution of human walking bouts during daily living, plotted as percentage of observed bouts vs. number of steps (in bins of ±1), as reported by *Orendurff et al., 2008*. About 50% of bouts were less than 16 steps (shaded area), observed from ten adults over fourteen days. (**C**) A typical walking task is to walk a given distance $D$, starting and ending at rest. (**D**) Walking speed is therefore expected to be a trajectory that starts and ends at zero, potentially differs from steady $v^*$, and has a finite duration $T$. Hypothetical trajectories are shown as dashed lines.

these everyday observations. Realistic walking tasks must therefore be governed by more than energy economy alone.

The specific energy measure thought to govern steady walking speed is the gross metabolic cost of transport (COT). Defined as energy expended per distance travelled and body weight (or mass), it has a convex dependency on speed. Its minimum (termed min-COT here) seems to predict the steady preferred speed of about 1.25 ms$^{-1}$, as widely reported for laboratory settings (*Ralston, 1958*; *Martin et al., 1992*; *Willis et al., 2005*; *Browning and Kram, 2005*; *Browning et al., 2006*; *Rose et al., 2006*; *Entin et al., 2010*), and observed of some other animals (*Hoyt and Taylor, 1981*; *Langman et al., 2012*; *Watson et al., 2011*). However, much of actual daily living also involves relatively short bouts of walking (*Figure 1B*), with about half of daily bouts taking less than 16 steps as reported by *Orendurff et al., 2008*. Such bouts, say of distance $D$ (*Figure 1C*), may spend substantial time and energy on starting from and stopping at rest, and relatively little time at steady speed. For example, in short bouts of walking up to about a dozen steps, peak speed is slower than the steady optimum, and only attains that value with more steps (*Seethapathi and Srinivasan, 2015*). There is a substantial energetic cost to changing speeds that could account for 4–8% of daily walking energy budget (*Seethapathi and Srinivasan, 2015*). If energy economy is important for walking, it should apply to an entire walking bout or task, and not only to steady speed.

Another important factor for walking is time. Time is valuable in part because energy is always expended even when one is at rest (*Jetté et al., 1990*), and because walking faster can save time to reach destination, but at greater energy cost (*Ralston, 1958*). Time is also subjectively valuable, because the urgency of a task, or even of an individual's personality, surroundings, or culture, could influence their speed. It has long been observed that people walk faster in big cities than in small towns, by a factor of more than twofold (about 0.75–1.75 ms$^{-1}$), or about ±40% of 1.25 ms$^{-1}$ (*Bornstein and Bornstein, 1976*). Perhaps population density affects a person's valuation of time (*Bornstein, 1979*; *Levine and Bartlett, 1984*; *Li and Cao, 2019*). Time is certainly a factor in deciding whether to walk or run (*Summerside et al., 2018*), and is considered an important factor in the general vigor of movements, beyond walking alone (*Labaune et al., 2020*). It is clearly worthwhile to expend more energy if time is of essence.

There are, however, challenges to incorporating time into walking. One method is to factor time into the equivalent of temporally discounted reward (*Shadmehr et al., 2010*), which refers to offering a reduced reward for longer durations, typically employed in fields such as movement vigor, foraging theory (*Green and Myerson, 1996*), and reinforcement learning (*Sutton and Barto, 2018*). Another is to express time as a cost that increases for longer movement durations, trading off against greater energy cost for shorter durations. Both the energy cost for an entire walking bout, plus a cost for time duration, could thus be combined into a single objective function to be minimized (*Wong et al., 2021*). This presents a second challenge, which is how to determine the optimum. Unlike the case of steady walking at a single speed (*Figure 1A*), an entire walking bout requires a time-varying trajectory of walking speed. This cannot be determined from the cost of transport curve, but can potentially be predicted by a quantitative, dynamical model. Simple models of walking (Figure 7), based on the pendulum-like dynamics of walking, can predict aspects such as optimal step length and step width (*Kuo et al., 2005*) for a steady speed, and optimal speed fluctuations for uneven steps (*Darici and Kuo, 2022a*; *Darici and Kuo, 2022b*). It remains to be determined whether they can predict the energetics and timing of walking bouts with transient conditions.

The purpose of the present study was to test whether the combined costs of energy and time can predict dynamic variation in walking speed. We propose a basic quantitative objective function called the Energy-Time hypothesis, which includes a cost for total energy expenditure or mechanical work for a walking bout, plus a penalty increasing with the bout's time duration. We apply this objective to a computational walking model, using dynamic optimization to predict dynamic speed profiles for walking bouts of varying distance (*Figure 1D*). For relatively short walking bouts, this hypothesis predicts speeds that vary within a bout, and speed profiles that vary across bout distances. For longer distances, it predicts a steady walking speed, not as an explicit outcome but rather as an emergent behavior. To test these predictions, we performed a human subjects experiment, comparing empirical speed profiles against model predictions. If the model is able to predict human speed profiles, it may suggest that a valuation of time and energy can influence walking, and thus be compatible with walking bouts of any distance and any degree of urgency.

## Results
### Model predictions

A simple model of walking dynamics predicts theoretically optimal speed trajectories and walking bout durations. The Energy-Time hypothesis is that humans perform walking bouts that minimize an objective including the total energy and time expended for the bout. The dynamic optimization problem may be summarized as

$$\text{minimize (Energy expenditure)} + C_T(\text{Time duration})$$

$$\text{subject to : starting and ending at rest}$$

$$\text{with } N \text{ steps of pendulum} - \text{like walking dynamics at human} - \text{like step length}$$

where the total metabolic energy expenditure is evaluated for the entire walking task, and the time duration is weighted by a metabolic energy coefficient $C_T$ (in units of energy per time). In the model, positive mechanical work is used as a proportional indicator of human energy expenditure, with (lower-case) work coefficient $c_T$. This coefficient is a valuation of time, and may be interpreted as the energy

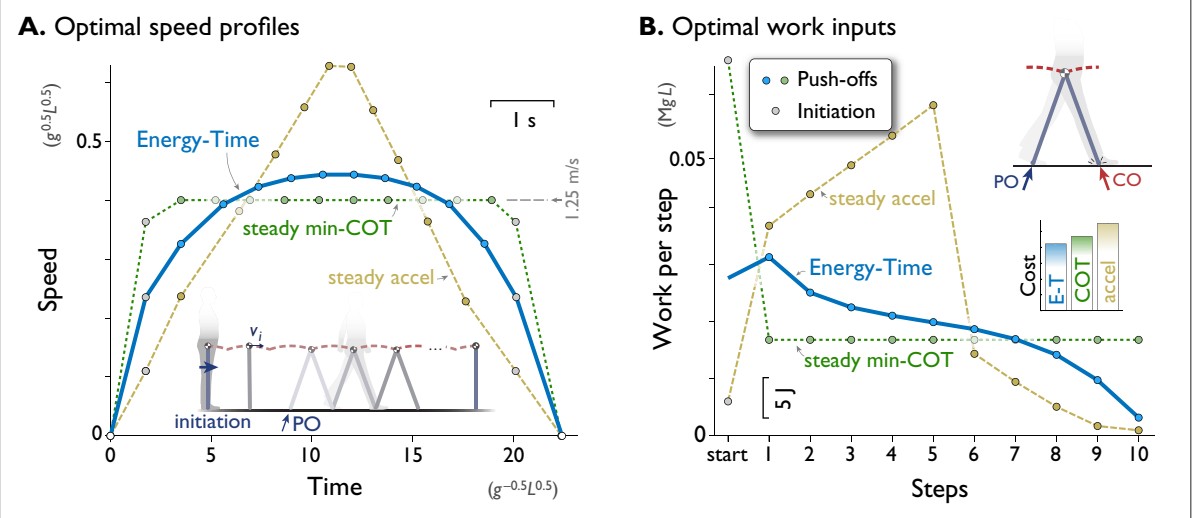

**Figure 2.** A computational model of walking predicts that a rounded speed profile is most economical for a short walking bout of fixed time and distance. (**A**) Predicted speed profiles for a walking bout of ten steps, comparing minimization of Energy and Time (Energy-Time, solid line) against maintaining steady speed (min-COT, dotted line) or steady acceleration and deceleration (steady accel, dashed line). Energy-Time minimizes the total push-off (PO) work plus time expended for a walking bout, for a model with pendulum-like legs (inset). Steady min-COT walks at the steady speed that minimizes cost of transport, by accelerating immediately to that speed. Steady acceleration walks with linearly increasing speed until mid-point, then decelerates linearly back to rest. Energy-Time predicts a gently rounded speed profile, min-COT a trapezoidal profile (always at min-COT speed), and steady acceleration a triangular profile. Speeds are discretely sampled as the average forward speed over each step (filled dots), starting with an initiation impulse to accelerate from standing and a termination impulse to decelerate at the end (gray dots). (**B**) Positive work inputs for each hypothesis, including initiation work (gray dots) and push-off work (colored dots, one per step). Energy-time hypothesis predicts the least total work (inset bar graph compares Energy-Time "E-T", min-"COT", steady-"accel" costs). Predictions are for a dynamic walking model with pendulum-like legs (inset, see Methods). All predictions are designed for the same duration based on steady min-COT speed as a reference, resulting in cost of Time $c_T = 0.020$. Predictions are plotted in terms of normalized units based on body mass $M$, leg length $L$, and gravitational acceleration $g$; scale for typical human also shown, mass 70 kg, leg length 1 m.

or work one is willing to expend to save a unit of time. The overall objective is to be minimized with an appropriate trajectory of the model's speed, which in the human is the outcome of active control actions. The optimal control actions are subject to constraints, namely the specified distance of a walking bout and the governing walking dynamics (see Methods for details). Walking dynamics refers to the dynamics of the body, where the stance leg behaves like an inverted pendulum and the swing leg like a swinging pendulum. These dynamics also describe the mechanical work and energy associated with a speed trajectory, and how long each step takes. Step length was nominally kept fixed, and then varied in parameter sensitivity studies below. The time duration $T$ of a bout is the outcome of the optimization, where greater valuation of time $C_T$ favors shorter duration.

The optimization predicts the speed profiles for a representative, ten-step task (***Figure 2A***). To focus on Energy first, the duration is kept fixed here. The Energy-Time objective predicts a gradual increase in speed, with a gently rounded profile that peaks mid-way through the bout. For this relatively short distance, little or no time is spent at steady speed. This contrasts with two other possibilities, to maintain steady speed at min-COT (***Ralston, 1958***), or to maintain steady acceleration. The steady min-COT objective produces a speed profile resembling a trapezoid, accelerating immediately to attain a fixed steady speed, maintained throughout the bout, before terminating just as quickly. Steady acceleration causes speed to increase linearly over time until peaking mid-bout, followed by a linear decrease back to rest. Here, all three alternatives are directed to walk the same distance in the same time, but at different costs.

Examination of the positive work inputs reveals why Energy-Time is least costly (***Figure 2B***). Its gentle acceleration requires moderate push-offs, which trail off over time as the model nearly coasts to a stop at destination, taking advantage of each step's collision loss to reduce speed at little cost. In contrast, the steady min-COT objective pays a high cost to initiate gait, and then a moderate and constant amount of work for all push-offs. It also does not take advantage of coasting to a stop. Steady acceleration pays a high cost to peak at a high speed, which is not made up for by greatly

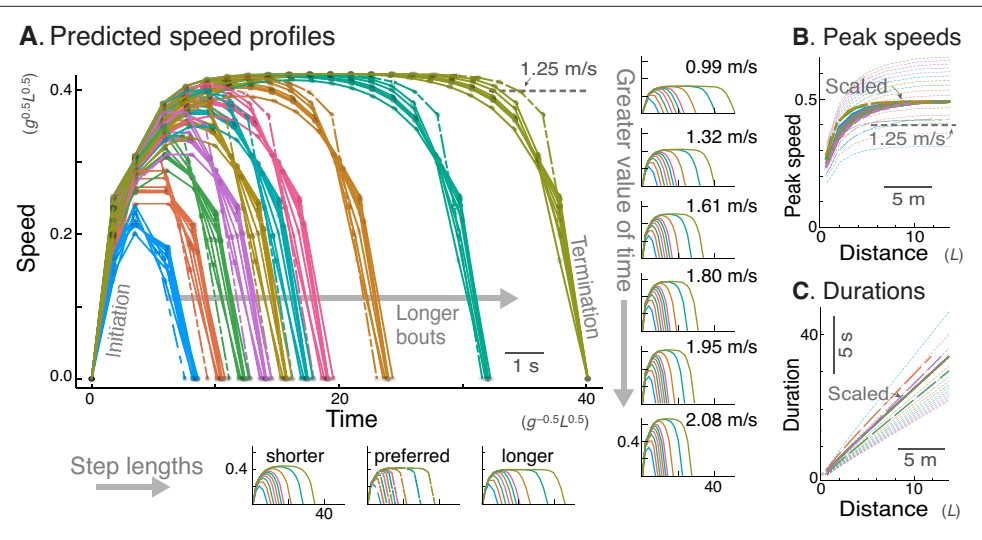

**Figure 3.** Energy-Time hypothesis predicts a family of speed profiles. (**A**) Predicted speed profiles vs. time for a range of walking distances, with longer bouts reaching higher and steadier peak speeds. In the main plot, multiple predictions for different time valuations and step lengths are scaled and superimposed on each other to emphasize self-similarity. Original, unscaled traces are shown in surrounding insets. (Horizontal insets, bottom:) Three different step lengths including shorter and longer steps than nominal, and human preferred step length relationship (dashed lines); main plot also includes nominal step length. (Vertical insets, right:) Varying valuation of time $c_T$ results in two-fold variation in peak speeds (labeled) and walk durations. The time cost and step length therefore affect only how quickly the task is completed, and not the shape of the family of speed profiles. Shown are trajectories of discrete speed, defined as the average forward speed over each step. (**B**) Peak speeds are predicted to increase sharply with distance, approaching an asymptote for distances of about 12 m or more. Again, despite different peak speeds, the curves are self-similar and can be scaled to a single shape (thick lines). (**C**) Walking durations increase with distance, with slightly curvilinear relationship (also scaled to a single shape, thick lines). In (**A**), time cost $c_T$ is varied between 0.006 and 0.06 (in units of $Mg^{1.5}L^{0.5}$), and distances range from 2 to 20 steps. Model predictions are plotted in dimensionless units, using body mass $M$, leg length $L$, and gravitational acceleration $g$ as base units; scale for typical human also shown, mass 70 kg, leg length 1 m.

reduced push-offs as it comes to a stop. Some intuition may be gained by considering the analogous situation of a vehicle driving a short fixed distance between two stop signs, in fixed time. It is generally economical to accelerate and decelerate gradually, and not necessarily maintain steady speed except beyond a certain distance. A trapezoidal (min-COT) speed profile is not economical, because considerable energy is spent in fast acceleration, and braking maximally at the end is more wasteful than lifting off the accelerator early and coasting. A triangular (steady acceleration) profile is also not favorable, due to the work needed to briefly attain a high speed. Of course, walking and driving have different dynamics, but both have similar energetic loss rates that increase approximately with the cube of speed. The higher losses incurred at greater speeds is an important reason for the Energy-time optimality of a rounded speed profile. For this task, the min-COT hypothesis ultimately costs 11% more total work, and the steady acceleration hypothesis 31% more, than minimizing Energy-Time.

Having established the energetic advantages of the Energy-Time hypothesis, we next examine how the optimal speed profiles vary with Time and other model parameters (*Figure 3*). Here, there are three parameters of interest: the value of time $c_T$, step length, and walking bout distance. We considered step lengths $s$ fixed at nominal (0.68 m), at slightly shorter and longer lengths (0.59 m and 0.78 m), and increasing with speed according to the human preferred step length relationship (see Methods for details). We also considered bouts of one to twenty steps, or about 0.68 m–13.7 m, as well as time valuations $c_T$ ranging ten-fold, 0.006–0.06 (dimensionless). Regardless of the combination of these parameters, a few characteristics emerge. The speed profiles generally retain a gently rounded profile (*Figure 3A*), smoothly accelerating from rest and leveling off at a peak speed before decelerating back to rest. Unlike the trapezoidal, min-COT prediction, the human speed profiles are always peaked, particularly for short bouts. The longer the distance, the greater the peak speed (unlike min-COT), and the more sustained that peak, contrast to more

rounded speed profiles for shorter distances. The acceleration and deceleration slopes increase slightly with longer bouts, and only for distances of about 10 m or more is there a steady gait near peak speed. The peak speed also initially increases sharply with walking distance (*Figure 3B*), but then approaches an asymptote for greater distances, as the cost of acceleration and deceleration becomes inconsequential to overall cost $J$. In fact, the asymptotic peak speed for long walks is a steady speed, not unlike the minimum-COT speed. But for finite walk distances, the speed profile generally does not agree with the steady min-COT hypothesis, because it varies with bout distance, and dynamically within each bout.

Another feature of the Energy-Time prediction is consistency with respect to parameter values (*Figure 3A*). The main free parameter is the time valuation $c_T$, for which higher values call for higher peak speeds, and therefore shorter walking durations. But with peak speeds ranging more than two-fold (*Figure 3A*, inset), the speed profiles all had similar shape. In fact, scaling each of the profiles in time and amplitude yielded a very similar family of trajectories regardless of parameter values (*Figure 3A*). This is also the case for variation in step length, with nominal, long, and preferred human step lengths all producing similar trajectories. Similarly, the peak speed vs. distance curves resembled a saturating exponential regardless of parameter values (*Figure 3B*), and these were also scalable in amplitude to yield a single family of curves. Walking durations vs. distance (*Figure 3C*), also had similar, scalable and curvilinear shape for all parameters. Similar profiles are produced regardless of whether the model takes step lengths that are fixed, or that scale according to the empirical step length vs. walking speed relationship for steady walking (insets, *Figure 3A*). We therefore subsequently keep step length fixed (equivalent of 0.68 m for human) for simplicity. As a result, the time cost coefficient $c_T$ is effectively the model's sole free parameter, and the predicted speed profile shapes scale very consistently with respect to that parameter.

There are thus three main predictions from the model that can be tested in human. First, the speed profiles should fall within a single consistent family, which includes more rounded shapes for short walks, and flatter for longer walks (*Figure 3A*). These profiles should exhibit self-similarity, and be scalable in peak speed and time to resemble a single, relatively uniform family of profiles. Second, the peak speed should increase with distance, with an approximately exponential saturation toward an asymptote (*Figure 3B*). Again, that relationship is expected to be scalable by peak speed, and testable by a single saturating exponential. And third, walking durations should increase with distance, in a slightly curvilinear relationship (*Figure 3C*) approaching a straight-line asymptote for longer distances. For shorter distances, much of the time should be spent accelerating and decelerating, compared to relatively brief cruising periods that become proportionately greater with distance. We thus treat the time valuation $c_T$ as an empirical parameter that mainly affects the scale, but not the shape of the speed profiles and dependency on distance.

## Experimental results

The human speed profiles for all trials and all distances were found to exhibit consistent profiles between subjects and between individual trials (*Figure 4*). These profiles resembled predictions from the Energy-Time hypothesis. Qualitatively, humans produced inverted U profiles similar to model, with sharper and lower peak speeds for shorter bouts. Longer bouts had higher and flatter peaks, where a steady speed could be discerned. Each individual subject walked at a somewhat different speed and for a somewhat different time (*Figure 4A*). For example, the range of peak speeds across subjects, observed for the longest (12.7 m) bout, was 1.21 $\text{ms}^{-1}$.78 $\text{ms}_{-1}$, and the corresponding range of durations was 8.51–11.86 s. Nevertheless, the profile shapes were all quite similar across subjects. In contrast to the min-COT hypothesis, the human peak speeds increased with distance, many well below the min-COT speed of about 1.25 $\text{ms}^{-1}$. In addition, the human speed trajectories did not resemble the trapezoidal profiles of the steady min-COT hypothesis for all distances, nor the triangular profiles of steady acceleration.

The experimental speed trajectories were scalable in speed and time, to yield a self-similar family of trajectories (*Figure 4B*). Each individual's trajectories were normalized in time by the duration for that subject's longest bout, and in speed by the maximum speed of their longest bout. These were then re-scaled to match the average duration and peak seed across subjects, to yield a normalized set of speed profiles for all subjects (*Figure 4B*). The resulting normalized trajectories reveals considerable similarity between individuals, with a single, relatively uniform family of profiles for all subjects.

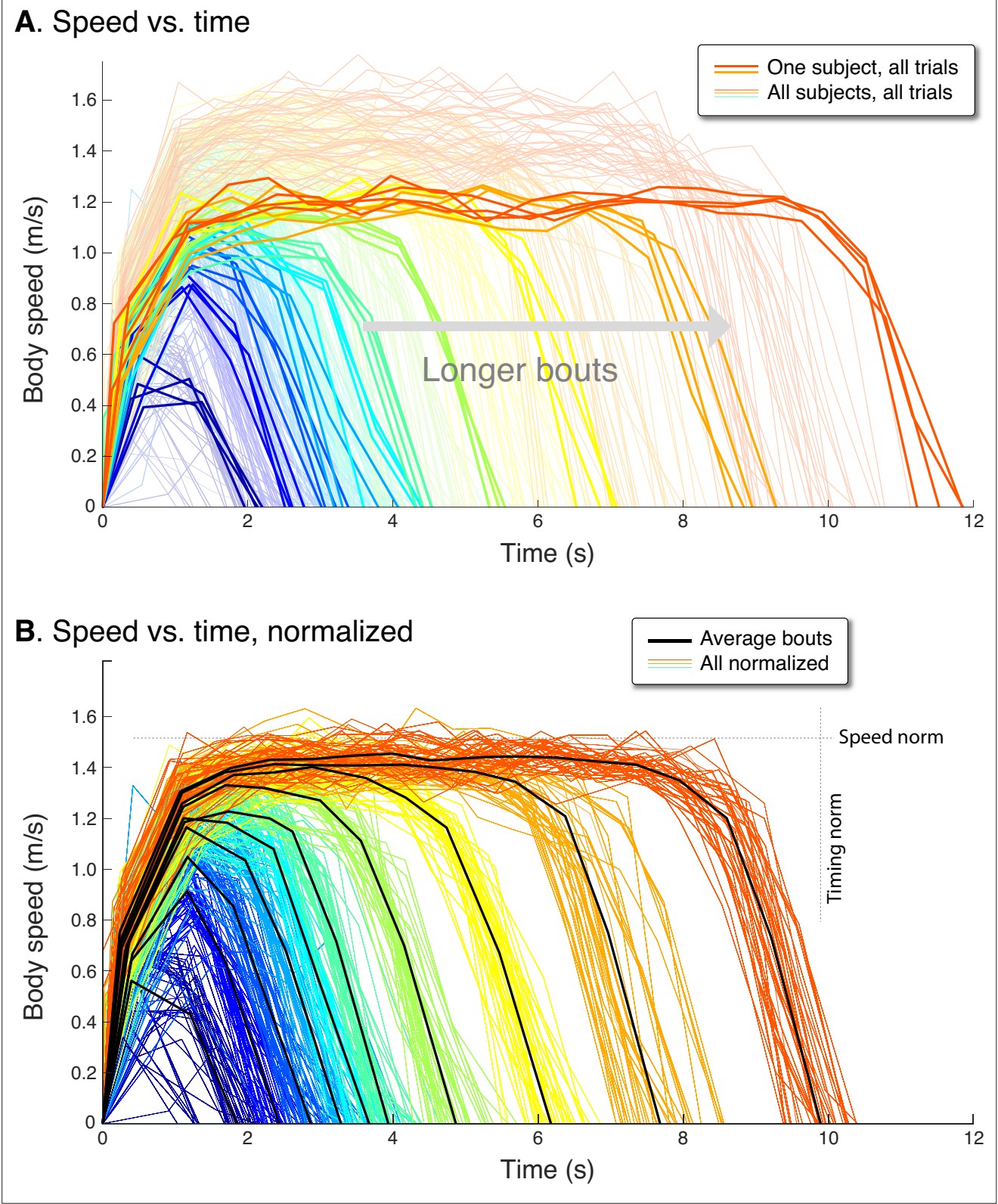

**Figure 4.** Human speed profiles vs.time for (**A**) all subjects ($N = 10$), and (**B**) all subjects normalized to the average. Body speeds are plotted for all ten walking bout distances (colored lines). In (**A**), one representative subject is highlighted (thicker lines) to show a typical person's variability between trials. In (**B**), all traces for each subject are normalized by that person's average peak speed ('Speed norm') for the longest distance, and by their average time for that bout ('Timing norm'). Also shown are the mean walking bouts across subjects (thick black lines, $N = 10$) for each distance, to illustrate

*Figure 4 continued on next page*

*Figure 4 continued*

how different subjects resemble each other despite varying in how fast they walk. Average bouts were computed by resampling each trial to the most common step count for each distance, averaging across such profiles for each distance, and then rescaling time to reflect the average duration for each distance. Body speed is defined (in Methods) as an average for each step, dividing step length by the time between between mid-stance instances.

Thus, the peak speed and duration of a walking bout of 2 m was consistently related to one of 12 m, and vice versa.

This scalability may be quantified in terms of peak speeds and durations. Examining the peak speed for each distance reveals a consistent pattern (*Figure 5A*). Peak speeds increased with distance, sharply for short distances and then saturating for longer distances. The overall pattern resembled a saturating exponential, similar to model predictions. The overall maximum speed was $1.52 \pm 0.14 \, \mathrm{ms}^{-1}$ (mean ± s.d. across subjects), almost always for the longest distance. We normalized each individual's peak speed by their own maximum, and found the resulting peak speed vs. distance curves to be scalable into a single normalized curve across subjects. With normalization, the variability (s.d. across individuals) of peak speeds was reduced significantly ($P = 1.6 \times 10^{-6}$), by $0.07 \pm 0.02 \, \mathrm{m \, s}^{-1}$ (mean ± s.d.) across all bout distances or about 54% compared to un-normalized. Thus, even though each individual walked at their own pace, that tendency was consistent across all distances. Much of the inter-subject variability was reduced by normalizing the peak speeds, revealing a common relationship between peak speed and bout distance.

There was a similarly consistent pattern for walking durations across distances (*Figure 5B*). Walking durations increased with distance in a slightly curvilinear fashion. Again, we normalized each individual's durations by the duration for the longest bout ($9.86 \pm 0.75 \, \mathrm{s}$), and found the duration vs. distances to be scalable into a single normalized curve across subjects. With normalization, the variability of durations was also reduced significantly ($P = 0.03$), by $0.10 \pm 0.13 \, \mathrm{s}$ across all bout distances, or about 18% compared to un-normalized. Similar to peak speeds, much of the inter-subject variability was reduced by normalizing. There was a common and consistent relationship between different walking bouts, similar to model predictions.

The change in peakiness or flatness of speed profiles was indicated by the time spent accelerating, decelerating, or at approximately constant speed (*Figure 5B*). This was described by rise time, defined as the time to accelerate from 0% to 90% of peak speed, cruise time as the time spent at 90% of peak speed or more, and fall time as the time to decelerate between 90% and 0% of peak speed (*Figure 5B*). These measures of time increased with bout distance. As a fraction of each bout's duration, the rise and fall times appeared to take up a greater proportion for shorter bouts, and only a very small proportion was spent at steady speed. Conversely, cruise time took up a greater proportion of the time for longer bouts. These behaviors were consistent with predictions from the Energy-Time hypothesis.

The peak speed was described reasonably well by a saturating exponential (*Figure 5A*). An ad hoc, least-squares nonlinear fit to the normalized data yielded a saturating exponential curve

$$v(D) = c_v(1 - e^{-D/d_v}) \tag{1}$$

where $v(D)$ is the peak speed as a function of total walking distance $D$, and fitted values were $c_v = 1.516 \, \mathrm{m \, s}^{-1}$ (1.496, 1.536 CI, 95% confidence interval) and $d_v = 1.877 \, \mathrm{m}$ (1.798, 1.955 CI), with a goodness-of-fit of $R^2 = 0.86$. The curve fit shows that there was considerable consistency in maximum speed; even short walking bouts of slow peak speed were still consistent with longer bouts of higher speed.

Similarly, walking duration increased with walking distance (*Figure 5B*), with a slightly curvilinear relationship. The total walk duration $T(D)$ may be treated as a saturating exponential approaching a straight asymptote, defined ad hoc as distance $D$ divided by preferred steady walking speed plus an offset $T_0$. The curve was of the form

$$T(D) = \frac{D}{v_T} + T_0(1 - e^{-D/d_T}). \tag{2}$$

where fitted coefficients were $v_T = 1.494 \, \mathrm{m \, s}^{-1}$ (1.466, 1.521 CI), $T_0 = 1.470 \, \mathrm{s}$ (1.375, 1.565 CI), and $d_T = 0.790 \, \mathrm{m}$ (0.610, 0.970 CI).

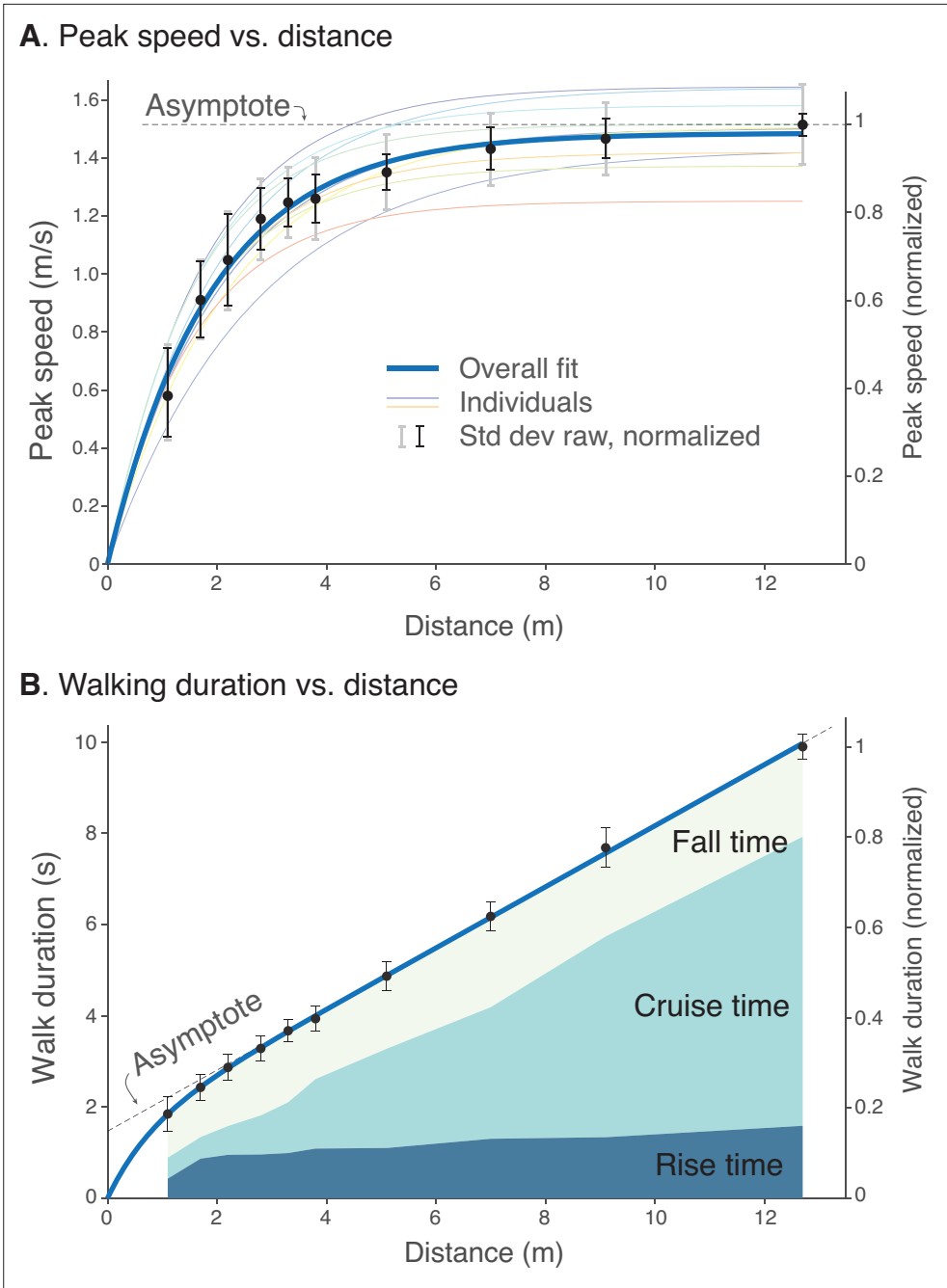

**Figure 5.** Human walking bouts show increases in (**A**) peak speed and (**B**) walking duration vs. distance.
(**A**) Peak speeds are shown for each walking distance, averaged across subjects' normalized data (filled symbols, $N = 10$), along with variability (standard deviation, thin black error bars). These are accompanied by a saturating exponential fit (thick solid line, $R^2 = 0.86$). Also shown are fits for each subject's unnormalized data (thin colored lines; dimensional vertical axis at left), and unnormalized variability between all trials (standard deviation, light gray error bars). (**B**) Walking durations are shown for each walking distance, averaged across subjects' normalized data (filled symbols), along with variability (standard deviation, error bars) and a saturating exponential fit ($R^2 = 0.98$). Shaded areas denote rise time (0%–90% of peak speed), cruise time (90% of peak and greater), and fall time (90%–0%). Rise and fall times appear to dominate shorter walking bouts, and cruise time for longer walking bouts. Filled black dots denote mean data, error bars denote s.d. The entire range of unnormalized peak speeds and durations for all subjects is shown in *Figure 4*, along with normalization definitions (Speed norm, Time norm). Dimensional vertical axis (left) is based on the average normalization constant relative to normalized data (right vertical axis). Normalization yielded reduced variability in peak speeds and durations.

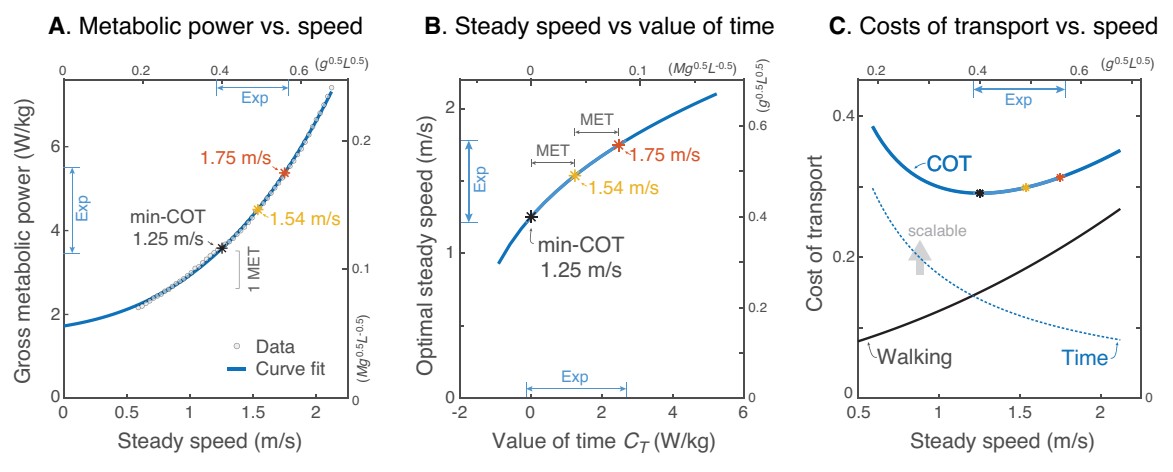

**Figure 6.** Prediction of steady walking speed emerges from Energy-Time hypothesis. (**A**) Human metabolic power vs. speed for steady walking (adapted from Fig. 11 of **Elftman, 1966**), along with a model-based curve fit ($R^2 = 0.999$; see **Equation 15**). Faster walking can be produced by valuing time more, with metabolic $C_T = 1\,\text{MET}$ yielding 1.54 ms⁻¹ and $2\,\text{MET}$ yielding 1.75 ms⁻¹. (MET is metabolic resting rate, serving as a standard reference value.) (**B**) Model steady speed vs. value of time $C_T$ increases such that each increment of $C_T$ in model yields a diminishing increase in speed, due to the increasingly high energetic cost of walking faster. Walking speeds observed in present experiment ('Exp' range) may be interpreted as human $C_T$ ranging from about 0–2 MET above resting. (**C**) Model energetic cost of transport (COT) may be regarded as the sum of two competing costs: a net physiological cost for Walking and a scalable cost for the Time expended. Steady walking speed is optimized where the two costs have equal and opposite slope. As the valuation of time $C_T$ increases, preferred steady speed increases. The valuation of time includes the resting metabolic rate plus a subjective component that does not literally cost energy. The valuation of time represents how much metabolic energy an individual is willing to spend to save a unit of time. The three marked speeds (asterisks '*') roughly denote the minimum, mean, and maximum steady speeds observed here; they correspond with a gross valuation of time starting at resting rate, and incremented by one or two MET.

We also performed similar analyses on a grass walking surface to test for sensitivity to slightly uneven terrain. An identical set of conditions was collected on short grass outdoors. The fit of peak speed vs. bout distance yielded $c_v = 1.446$ ms⁻¹, $d_v = 1.822$ ms⁻¹ ($R^2 = 0.85$), and for duration versus distance $v_T = 1.426$ ms⁻¹, $T_0 = 1.336$ s, and $d_T = 0.503$ m ($R^2 = 0.98$). These relationships were quite similar to those obtained on sidewalk.

We next estimated the relationship between human valuation of time and steady walking speed (**Figure 6**). Here we used an empirical human metabolic power curve (**Figure 6A**) from literature (**Elftman, 1966**) to predict how steady walking speed should increase with metabolic value of time $C_T$ (**Figure 6B**), and how the energetic cost of transport vs. steady walking speed (**Figure 6C**) may be regarded in terms of competing costs for Energy and Time. This human power curve was fitted to the model, to facilitate scaling the model's mechanical energy into human metabolic energy. The optimal steady walking speed emerges from that curve (**Srinivasan, 2009**), as a function of $C_T$ (**Figure 6A**). Based on these crude assumptions, a time valuation of zero yields the same optimal speed $v^*$ of 1.25 ms⁻¹ as min COT, and close to the minimum steady speed (among subjects) of 1.21 ms⁻¹ observed here. It is instructive to increment $C_T$ by multiples of the metabolic equivalent (MET), a standard physiological resting rate of about 1.23 Wkg⁻¹ [Watts per kilogram] (**Jetté et al., 1990**). An increment of 1 or 2 MET yields optimal speeds of 1.54 ms⁻¹ or 1.75 ms⁻¹, respectively, quite close to the observed mean and maximum steady speeds (among subjects), respectively (of 1.52 ms⁻¹ and 1.75 ms⁻¹). Thus, if the same metabolic power curve were applicable to all, the slowest subject would have valued time at about zero MET, the mean subject at +1 MET, and the fastest at +2 MET. This also suggests that most subjects preferred faster steady speeds than min-COT. Incrementing $C_T$ yields diminishing returns in speed (**Figure 6B**), because it is increasingly costly to walk faster. One interpretation afforded by the Energy-Time hypothesis is that there is an effective cost of transport that may be separated into two terms (**Figure 6C**), Walking and Time: one for the net metabolic cost for walking alone (due to push-off work), the other a cost of time that lumps the resting rate with (or within) an individual's $C_T$. This reveals a trade-off, where the cost of walking increases with speed, and the cost of time decreases (hyperbolically with speed), such that the two opposing curves (or rather their opposing slopes) determine an optimum. A greater valuation of time adds to this effective cost of transport,

equal to the actual metabolic energy plus a subjectively scalable cost of time, per distance traveled. This shows how the effective Energy-Time cost per distance is minimized at higher speeds for greater $C_T$.

## Discussion

We had sought to test whether humans optimize not only metabolic energy but also a valuation of time spent walking. Although the prevailing theory of minimizing the energetic COT explains steady walking, it does not explain shorter walks that lack a steady speed, nor does it readily accommodate individual tendencies toward faster or slower speeds. We found that humans walk bouts of finite distance with a trajectory of speeds varying with distance. These bouts fall within a consistent family of trajectories across subjects, despite individual differences in overall speed or duration. These results are in agreement with a simple mechanistic model of walking, governed by optimization. The findings suggest that humans optimize a combined objective that trades off the energy to arrive at destination against the time it takes to get there.

Each human walking bout consisted of a dynamically varying trajectory of speed with an inverted U shape. Many of these bouts included a period of steady walking, at speed similar to (but usually faster than) the supposed min-COT (*Ralston, 1958*), but mainly for the longer distances (*Figure 4*). Shorter bouts of say 10 m or less exhibited a relatively brief peak slower than the typical min-COT speed (*Figure 5B*). All such bouts also spent substantial time and energy in acceleration and deceleration (*Figure 5B*). Moreover, short distances such as this are quite ecological, accounting for about half of the daily living walking bouts reported by *Orendurff et al., 2008*, with acceleration and deceleration potentially accounting for 4–8% of daily walking energy budget (*Seethapathi and Srinivasan, 2015*). This contrasts with the min-COT hypothesis, which predicts only a single steady speed, and cannot explain how to start or end a bout. People usually walk to a destination of known and fixed distance, for which it is more sensible to minimize the total energy for that distance, rather than energy per distance (min-COT). This is not to dismiss the energy spent for steady walking, which has been well-characterized in laboratory settings where the vast majority of published studies have been performed. But in daily living, humans walk a variety of speeds and distances, many too short to be steady. To our knowledge, this is the first study to consider the time and energy spent in brief walking bouts.

Even though there were considerable differences between individuals, each subject was quite consistent within their own walking bouts. Those with a slower or faster peak speed during longer bouts were also consistently so during shorter, non-steady bouts (*Figure 5*), as evidenced by the 54% reduced variability after normalizing peak speeds by the longest bout. Moreover, the bouts across all subjects were scalable to a single, self-similar family of trajectories (*Figure 4*). These trajectories were not consistent with a fixed acceleration or deceleration profile (*Figure 2A*), and instead exhibited a greater peak speed and longer time to that peak with greater distance (*Figure 4B*). This pattern suggests that there are systematic criteria or principles that govern walking bouts of finite distance. Even though some individuals are faster than others (*Figure 4A*), they all seem to follow similar principles.

These observations agree with the primary hypothesis that humans optimize for energy and time. The trade-off between the two, described by the valuation of time $C_T$ (for human metabolic cost, or $c_T$ in terms of model work), is readily explained for steady walking. A valuation of zero corresponds to minimizing the gross metabolic cost of transport (*Figure 6B*), yielding the min-COT speed. (Or equivalently, resting metabolic rate could be treated as part of a running time cost, trading off against the net metabolic rate for walking, as in *Figure 6C*.) Of course, humans do not always walk the same speed, and faster speeds save time but cost more energy to cover a given distance. One's actual steady speed must therefore be observed empirically, from which their valuation of time may also be estimated (ranging roughly 0–2 MET, *Figure 6B*). But what sets time valuation apart is its predictive value for non-steady walking. The speed trajectory for a fixed bout distance need not even contain a steady portion, and is often dominated by acceleration and deceleration. The Energy-Time model can nevertheless predict an entire family of trajectories across a variety of distances (*Figure 3*) despite individual-specific step lengths and time valuations. The model also suggests how peak speeds, and time to peak, and durations should increase with distance (*Figure 3*), similar to human data. All this is based on one free parameter, the individual-specific valuation of time. That valuation may depend on

complex physiological and socio-psychological traits, but it nonetheless appears to have predictive value for a given context. Not tested here is the presumption that different contexts, for example changing the saliency of a task or adding time pressure, will also lead to systematic changes in walking bouts. If an individual's valuation of time can be estimated empirically, our hypothesis provides an operational means of integrating it into a quantitative model.

These predictions are produced by a mechanistic model governed almost entirely by dynamics. The timing comes from the dynamics of pendulum-like walking, and the energetics from the step-to-step transition between pendulum-like steps. The step-to-step transition requires mechanical work to accelerate and to restore collision losses, such that for short walks it is uneconomical to accelerate quickly to min-COT speed (*Figure 2*, steady min-COT). An alternative is to accelerate more gradually, but that is costly because of the high peak speed attained (*Figure 2*, steady accel). The model thus favors an intermediate and smooth acceleration to a slower and continuously varying speed with an inverted U shape. Separate studies have found step-to-step transition work to predict human metabolic energy expenditure as a function of step length (*Donelan et al., 2002*) and changing speed (*Seethapathi and Srinivasan, 2015*). Here, we have constrained the pendulum-like dynamics so that there is only one physical parameter, step length, which in any case has very little effect on the characteristic shape of speed trajectories (*Figure 3*). As such, this was set to nominal value and not fitted to data, making the model as predictive as possible. Indeed, the very same model also predicts human compensation strategies for walking on surfaces of uneven height (*Darici and Kuo, 2022a*; *Darici and Kuo, 2022b*). Of course, the human body has many degrees of freedom capable of far different motions, but model analysis suggests that pendulum-like walking is the most economical means to move the COM at slow to moderate speeds (*Srinivasan and Ruina, 2006*), and that push-off during the step-to-step transition is the most economical means of powering such pendulum-like walking (*Kuo, 2001*). These models are predicated on mechanical work as the major cost, and the COM as the major inertia in the system.

It is instructive to consider what other models might explain or predict our experimental results. We did not explore more complex models here, but would expect similar predictions to result from any model based on pendulum-like walking and step-to-step transitions. This includes those cited in the previous paragraph, as well as a family of such models including three-dimensional motion (*Donelan et al., 2001*), knees and bi-articular actuators (*Dean and Kuo, 2009*), and plantarflexing ankles (*Zelik et al., 2014*). It is also possible that more complex, musculoskeletal models also perform substantial work and expend energy for step-to-step transitions, and might therefore agree with the present model. But here, relatively simple principles account for a fairly wide array of predictions (*Figure 3*), which are unlikely to result from happenstance. We are also unaware of any current hypothesis that could plausibly substitute for the present one. We therefore doubt if alternative models not based on pendulum-like principles could predict or reproduce these results, except with numerous fitted parameters.

This model is optimized with an additional control parameter, for the valuation of time. Time has long been recognized as a factor in the pace of life (*Levine and Bartlett, 1984*), and in reward and vigor in motor control (*Shadmehr et al., 2010*). It is typically expressed as a temporal discounting of reward, which appears key to human decision making and the theory of reinforcement learning. Here we expressed it as a trade-off equivalency between energy and time. This was mainly due to the need for compatibility with our energetics model, but also because neither model nor experiment included an explicit reward to be discounted. We used a simple, linear valuation of time in terms of energy, rather than a nonlinear, exponential or hyperbolic temporal discounting factor (*Green and Myerson, 1996*). Energy is a physiological cost endemic to life, that is not obviously more or less valuable at different points in time. It is sufficient to predict and explain the present results, and there is currently insufficient evidence to favor a nonlinear cost over our linear valuation. But regardless of the particular formulation, a default valuation of time may be an individualistic trait, generalizable to other tasks such as hand and eye movements (*Labaune et al., 2020*). Indeed, we have found a similar valuation of time to explain how reaching durations and speed trajectories vary with reaching distance (*Wong et al., 2021*). Another implication of our model is that humans may incorporate prediction of time within central nervous system internal models. Such models have long been proposed to explain humans predict and adapt their movement trajectories, for example to novel dynamics (*Todorov, 2004*). If movement duration is also part of human planning, it suggests the ability to predict not only

movement trajectories and energetics, but also time. Here such prediction is made operational within a quantitative model.

Valuation of time offers another perspective on minimizing the gross energetic cost of transport. Actual walking tasks are not purely steady, and are probably planned with consideration of what happens at the destination. *Long and Srinivasan, 2013* proposed a task to minimize the total energy expended to walk to destination within a more than ample allocation of time. They showed that total energy should be optimized by mixing resting and walking (and running if necessary). Suppose the task is extended to an indefinite duration, where a considerable amount of time is spent resting. The optimal total energy and walking duration may be found by applying our Energy-Time objective with time valuation ($C_T$) equal to zero (*Figure 6*). Walking faster than optimal would yield more time to rest, but at a greater total energy cost for walking. Walking slower would cost less energy for the walking motion alone, but at a greater total cost due to less time available to rest. After all, $C_T$ is the energy one is willing to expend to save a unit of time, and the resting rate is the energy expended to rest for a unit of time. This may seem like a trivial restatement of the min-COT hypothesis, but it differs in two important ways. First, it can predict both the duration of walking and the entire speed trajectory, even for short bouts where there is no steady portion. Second, it considers how valuable time is at the destination. Minimizing the gross cost of transport is most sensible for maximizing the survivable range distance (*Srinivasan, 2009*), which may not be a concern in modern life where survival rates are high, walks short, and calories plentiful. Rather, it may be a sensible default to value time at close to the resting rate (particularly for long walks), and then to vary the valuation dependingon context. One might thus rush toward a long-lost friend or hurry in a big city, if the time spent at destination is far more valuable than resting (*Bornstein, 1979*; *Levine and Bartlett, 1984*; *Li and Cao, 2019*). Similarly, we do not consider walking slowly to be a waste of energy per distance, but rather a waste of time. Even then, there are cases when humans might wish to waste time, for example to avoid an odious task, according to the expression 'the slow march to the gallows'.

The consistency of individual walking trajectories may have practical implications. Although walking speed is used as a clinical indicator of mobility, it is difficult to standardize (*Middleton et al., 2015*), because evaluations may be confined to the length of the available walkway, which may be too short (e.g. less than 10 m) for a steady speed to be reached. But given the time to walk a fixed distance, it may be possible to predict the duration and steady speed for another distance, referenced from a universal family of walking trajectories. We have identified one such family that applies to healthy young adults with pendulum-like gait. We do not know whether that family also applies to older adults, who prefer slower steady speeds and expend more energy to walk the same speed (*Malatesta et al., 2003*). Perhaps an age-related valuation of time might explain some of the differences in speed. Of course, some clinical conditions might be manifested by a deviance from that family, perhaps in the acceleration or deceleration phases, or in how the trajectories vary with distance. If quantified, such deviance might prove clinically useful. The methodology employed here does not require specialized equipment beyond inertial measurement units, and the characterization of distance-dependent speed trajectories can potentially provide more information than available from steady speed alone.

The Energy-Time hypothesis could be tested by further inquiries. We have thus far regarded the valuation of time as a relatively fixed parameter for each subject. That valuation is likely influenced, and therefore testable, by many contextual factors, including physiological and socio-psychological variables and task constraints. For example, caffeine intake, feeding status (e.g. *Taylor and Faisal, 2018*), or monetary reward could be used make time more valuable as a trade-off against energy. Conversely, energy may be helpful for assessing the valuation of time (or temporally discounted reward), which is not easy to measure other than indirectly. (Energy is also a universal currency, because all animals use energy, whereas only some use money.) Walking has a well-characterized physiological energy cost, and could serve as a useful trade-off against time or reward. The hypothesized optimal gait is the point at which the costs of energy and time have equal and opposite slopes (i.e., partial derivatives) with respect to an independent variable such as speed (e.g. *Figure 6C*), carried load, or incline. There are thus a variety of opportunities to manipulate the energetic cost of walking, as a means to assess the proposed valuation of time.

There are a number of limitations to this study. Although we tested model predictions in terms of speed trajectories, we did not measure mechanical work or metabolic energy expenditure in human subjects, which would provide greater insight regarding the proposed trade-offs against time. We

also did not evaluate each individual's metabolic cost of transport vs. speed, which would reveal more precise differences between the min-COT speed and the actual self-selected speed. Nor did we evaluate gait kinematics or kinetics, which may be helpful for detailing other ways that walking bouts vary with distance. The simple walking model also only includes a crude representation of step-to-step transitions, which we have crudely estimated to account for as much as 70% of net metabolic cost in steady walking (*Donelan et al., 2002*). We did not include other factors such as forced leg motion and step length modulation (*Doke et al., 2005*, up to say 33% of cost) that likely also affect energetic cost, and could therefore be used to test the valuation of time. Nor did we include factors such stability (*Bauby and Kuo, 2000*; *Donelan et al., 2004*; *Rebula et al., 2017*) and three-dimensional motion (*Donelan et al., 2001*), despite being part of our previous models, because we believe them to contribute little to the present task. In fact, because the optimal control model successfully completes the walking task, its feedforward motion attains implicit stability, which could reduce (but not eliminate) the need for feedback stabilization (*Darici and Kuo, 2022a*; *Darici and Kuo, 2022b*). We also did not include an explicit reward, which could facilitate assessment of energy and time in terms of other trade-offs such as money or food. In fact, the Energy-Time hypothesis should be regarded as a subset of the many factors that govern human actions, rendered here in a simple but quantitative form.

## Conclusion

Humans appear to select walking speed dynamically to minimize a combination of energy and time expenditure. This is both compatible with and extends the traditional hypothesis that humans minimize gross energy expenditure per unit distance. We found it more accurate to minimize the total cost of a walking bout, due to the ability to predict an entire speed trajectory, with the optimal steady speed as an emergent property. By including a cost for time expenditure, we introduce a quantitative and operational means to make walking models compatible with the study of movement vigor. Tasks may also be broadened beyond walking, to include consideration of the reward to be gained or further energy to be expended once the destination is reached. Walking may thus be integrated into broader questions of how and why humans take the actions they do. As a modification to the traditional adage about money, we suggest that 'Time is energy'.

## Methods

We experimentally tested how human walking speed varies with walking distance. The speed trajectories observed from human subjects were compared against predictions from the Energy-Time hypothesis and against the minimum-COT speed. To formulate the hypothesis and make quantitative predictions, we expressed it as an optimal control problem including both energy and time. We first

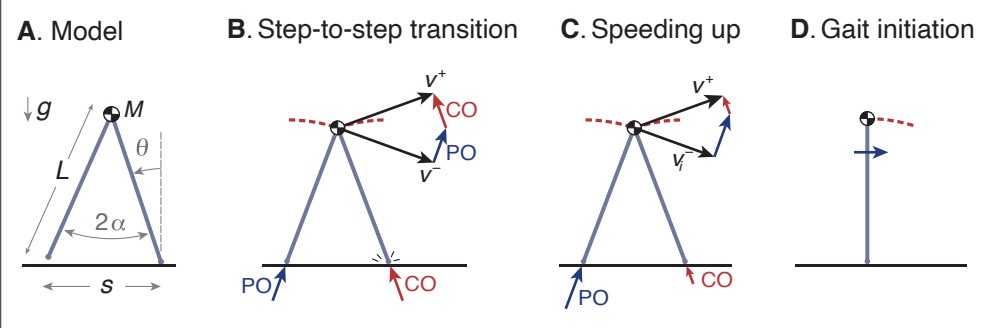

**Figure 7.** Simple optimization model of walking. (**A**) Walking dynamics modeled as a point center-of-mass (COM, mass $M$), supported by an inverted-pendulum stance leg (length $L$). (**B**) The inverted pendulum stance phase is punctuated by a step-to-step transition, modeled with an impulsive push-off (PO) from the trailing leg, followed by impulsive, inelastic collision (CO) with leading leg and ground. The COM velocity is $v^-$ at end of stance, then is redirected by PO and CO to yield velocity $v^+$ at end of step-to-step transition, beginning the next stance phase. (**C**) For the model to speed up, the magnitude of PO must exceed that of CO, and $v^+$ must have greater magnitude than $v^-$. (**D**) The walking bout is initiated by a forward impulse applied at the pelvis, described by positive work $u_0$.

state the hypothesis for human walking, and describe how it is adapted for a simple walking model to yield predicted speed trajectories. This is then followed by description of the experiment regarding human walking speed, and finally an analysis of steady speed as a property of the model.

## Walking model

We use the 'simplest walking model' (*Kuo, 2002*) to operationalize this optimization problem (*Figure 7A*). The model treats the stance leg as an inverted pendulum and requires mechanical work to power the gait. The body center of mass (COM) is modeled as a point mass supported by the stance leg, so that each pendulum-like step follows an arc, which itself requires no energy input. Work is performed during the step-to-step transition (*Figure 7B*), to redirect the COM velocity from forward-and-downward the end of one arc, and forward-and-upward at the beginning of the next. This is accomplished most economically with an active, impulsive push-off along the axis of the trailing leg, immediately followed by an impulsive, dissipative collision between the rigid leading leg and ground. In steady gait, the optimal push-off restores the collision losses, with mutually canceling impulses of equal magnitude. Speeding up is a matter of a greater push-off than collision, and a net increase in COM velocity during the step-to-step transition (*Figure 7C*). Positive and negative work are proportional to the square of the push-off and collision impulses, respectively (*Kuo, 2002*), so that speeding up also dissipates less collision energy than steady gait. Slowing down is the same in reverse, with collisions exceeding push-offs. This model predicts how step-to-step transition work for steady walking should increase as a function of step length and step width (*Donelan et al., 2002*). The model mainly predicts mechanical work for push-off, which appears to be a proportional predictor of the majority of human metabolic energy during steady walking (*Donelan et al., 2002*). That work also yields a mechanical cost of transport that varies curvilinearly with steady speed, similar to the empirical metabolic curve (*Figure 1a*; *Ralston, 1958*). There are of course other contributions to the metabolic cost of walking such as to move the swing leg (*Kuo, 2001*), but of smaller magnitude than step-to-step transitions, which are to be tested alone for their predictive value. Details of this model have been described in greater detail previously (*Darici et al., 2020*; *Kuo, 2002*), and are recounted only briefly here.

A walking bout consists of a sequence of $N$ steps, starting and ending at rest. It may be described by the discrete sequence of body speeds $v_i$ ($i = 1, 2, ..., N$), each equal to the distance traveled for step $i$ divided by that step's time duration $\tau_i$. The model begins at rest in an upright position (*Figure 7D*), and is set into motion by a forward initiation impulse acting on the pelvis. In humans, the torso can serve as an inertia that the hip muscles can act against, but for simplicity this action is represented as a translational impulse at the pelvis, summarized by the associated positive work $u_0$. The total positive work performed by the model consists of the work from initiation and the successive push-offs, a sequence $u_i$ ($i = 0, 1, ..., N$). There is also a corresponding sequence of dissipative collision impulses by the leading leg, and a dissipative gait termination to end at upright.

The step-to-step transition starts just before leading leg ground contact contact, and consists of a perfectly impulsive push-off from the trailing leg, followed in immediate succession by a perfectly inelastic and impulsive collision of the leading leg with ground. The COM velocity at the end of one stance phase is $v_i^-$, directed forward and downward according to the pendulum arc. Mechanical work is performed only during the step-to-step transition, with a succession of ideal impulses. First is positive push-off work from the trailing leg, directed from its foot to the COM, and second is a perfectly inelastic heel-strike collision of the leading leg with ground, directed from the leading foot to the COM. For brevity, the equations presented here use dimensionless versions of quantities, with body mass $M$, gravitational acceleration $g$, and leg length $L$ as base units. The push-off work is denoted $u_i$ (in units of mass-normalized work), and the push-off and collision sequence act to redirect the COM velocity to $v_i^+$ at the beginning of the next stance phase, directed forward and upward according to the next pendulum arc. Using impulse-momentum, the step-to-step transition is described by

$$v_i^+ = v_i^- \cos 2\alpha + \sqrt{2u_i} \sin 2\alpha \tag{3}$$

where $2\alpha$ is the inter-leg angle (*Figure 7A*). There is no work performed during the passive, inverted pendulum phases, and so the step-to-step transition is responsible for all energy inputs ($u_i$) and energy losses (from collisions).

The dynamics of an inverted pendulum describe all of the other motion in the system, consisting of the falling of one inverted pendulum toward the step-to-step transition, and the rising of the next inverted pendulum toward mid-stance. These dynamics determine the respective velocities and timing of these respective instances. The velocities may be found through conservation of energy:

$$v_i^- = \sqrt{2(1 - \cos \alpha) + v_i^2} \tag{4}$$

$$v_{i+1} = \sqrt{(v_i^+)^2 + 2(\cos \alpha - 1)} \tag{5}$$

The step time $\tau_i$ is defined as the time for the stance leg angle $\theta$ to move between successive mid-stance instants, and the corresponding velocities from $v_i$ to $v_{i+1}$. It may be regarded as the sum of a time $\tau_i^-$ from mid-stance to the step-to-step transition, and then the time $\tau_i^+$ from the step-to-step transition until next mid-stance. Using the linearized dynamics, the dimensionless time $\tau_i^-$ of step $i$ is

$$\tau_i^- = \log \frac{\alpha + \sqrt{v_i^2 + \alpha^2}}{v_i} \, . \tag{6}$$

The other time $\tau_i^+$ is

$$\tau_i^+ = \log \frac{\sqrt{v_i^+ + \alpha}}{\sqrt{v_i^+ - \alpha}} \tag{7}$$

For comparison with experiment, we also defined an average (as opposed to mid-stance) speed for each step $i$ as the step length divided by the step time between mid-stance instances,

$$\text{Body speed}_i = \frac{2L \sin \alpha}{\tau_i^- + \tau_i^+} \tag{8}$$

The trajectory of this body speed is plotted for different walking bouts, for both model and experiment. The equations for body speed and step time are summarized as constraints $f$ and $g$ below.

We chose nominal parameters to correspond to typical human walking. A person with body mass $M$ 70 kg and leg length $L$ of 1 m may typically walk at 1.25 ms$^{-1}$, with step length of 0.68 m and step time of 0.58 s, and corresponding fixed constant value $\alpha = 0.35$. The step length was also varied in parameter sensitivity studies. Using dynamic similarity, parameters and results are reported here either in SI units, or in normalized units with body mass $M$, gravitational acceleration $g$, and $L$ as base units.

## Optimal control formulation

We applied optimal control to the model for short walk bouts of varying distance (**Figure 7**). In humans, both positive and negative work appear to cost positive metabolic energy with different proportionalities (**Margaria, 1976**). In the model, we assess a cost only for positive work, because the net work of a level walking bout is zero. Minimizing positive work thus also implicitly minimizes the negative work, as well as metabolic cost of any proportionality. The push-offs have a one-to-one relationship with the speeds, and so either push-offs or speeds can can describe the trajectory. For the model, the goal is to minimize an objective function $J_{\text{model}}$ comprising the total positive work for the walking bout, plus the cost for the time duration:

$$J_{\text{model}} = (\text{Positive work}) + c_T(\text{Time duration}). \tag{9}$$

where the coefficient $c_T$ is the model's valuation of time in terms of work, and equal to the mechanical work the model is willing to spend to save a unit of time. It is treated as proportional to the human's valuation $C_T$ for metabolic energy per time.

This objective is applied as follows. The total distance $D$ of a walking bout may be achieved by taking an appropriate number of steps $N$. The walking trajectory is described by a discrete sequence of speeds $v_i$ (step $i = 1, 2, ..., N$), starting and ending from standing at rest, given a standard step length. The corresponding control actions include the initiation impulse and the push-off impulses, for a total of $N + 1$ actions $u_i$ ($i = 0, 1, 2, ..., N$). Using these variables, the model's objective is thus

$$J_{\text{model}} = \sum_{i=0}^{N} u_i + c_T \sum_{i=1}^{N} \tau_i \tag{10}$$

for the optimization problem

$$\underset{v_i\ (i=1,...,N)}{\text{minimize}} J_{\text{model}}(v_i) \text{ subject to} \tag{11}$$

$$\text{rest constraints} : v_0 = 0, v_N = 0 \tag{12}$$

$$\text{walking dynamics} : v_{i+1} = f(v_i, u_i), \tau_{i+1} = g(v_i, u_i). \tag{13}$$

where the model begins and ends at rest, and walking dynamics constrain how the speed and duration of the next step depend on the current step's speed and push-off (functions $f$ and $g$ detailed above). Note there are actually $N + 1$ controls, consisting of the initiation input $u_0$ and the $N$ step-to-step transition push-offs ($u_1, u_2, ...u_N$).

The time valuation $c_T$ is treated as an unknown but constant coefficient. Greater $c_T$ is expected to yield faster walking bouts, with experimental data used to determine an appropriate range of values. Within a fixed experimental context, we expect $c_T$ to be constant. We found values of $c_T$ ranging 0.006–0.06 $Mg^{1.5}L^{0.5}$ to yield speeds approximately similar to subjects.

Step lengths were examined with a nominal fixed step length of 0.68 m, and sensitivity analyses performed with fixed lengths of 0.59 m and 0.78 m, and varying lengths following the human preferred step length relationship $s = v^{0.42}$ (*Grieve, 1968*). We have previously proposed that step-to-step transitions account for most of the metabolic cost of human walking (say, up to 70% *Donelan et al., 2002*), and that forced swing leg motions to modulate step length also contribute a non-negligible (*Kuo, 2001*), but lesser cost (say, up to 33% *Doke et al., 2005*) that is neglected here. This is not to dismiss this and other costs of locomotion, but merely to hypothesize that step-to-step transitions will still dominate in transient walking bouts. The failure to include other costs, if sufficiently critical, should cause the model to make poor predictions.

We also considered two alternative hypotheses. One was that walking occurs almost entirely at the optimal steady speed. Termed the steady min-COT hypothesis, the goal is to walk at the min-COT speed $v^*$, or close to it, as much as possible. This is accomplished by minimizing deviations from $v^*$ throughout the bout, with objective

$$J_{\text{steady}} = \sum_{i=1}^{N}(v_i - v^*)^2 \tag{14}$$

subject to the same constraints as the Energy-Time hypothesis. This objective is expected to cause the model to accelerate immediately from rest to $v^*$, then remain at that steady speed, and then finally decelerate immediately back to rest. The second alternative was a steady acceleration hypothesis, which contrasts with min-COT's immediate high acceleration. Here, the acceleration is made as gradual as possible, albeit at the expense of a higher peak speed needed to travel the same distance and duration. Both of these alternatives help illustrate how different speed trajectories requires differing amounts of mechanical work, to be compared against the work produced by the Energy-Time hypothesis.

Model predictions were produced using computational optimization. Optimal control was computed using the JuMP optimization package for the Julia language (*Dunning et al., 2017*), formulated as a discrete collocation problem, minimized by nonlinear programming (Ipopt). Walking bouts were conducted for $N$ ranging 1–20 steps. The resulting trajectories were condensed into a scalable, self-similar family of speed trajectories.

## Experimental methods

We tested the model predictions by experimentally measuring the speed profiles of healthy adults walking a series of short distances, ranging about 2–20 steps. Subjects ($N = 10$, 6 male and 4 female, 24–38 yr) were instructed to walk at a comfortable speed in ten distance conditions, starting from standing at one numbered marker on the ground, and ending at another as requested by the experimenter. After each trial, there was a brief waiting interval of about 15–30 s, to reduce interference between successive trials and to avoid any incentive to rush through trials. The walking surface was a

level sidewalk. The numbered markers were separated by distances of 1.1, 1.7, 2.2, 2.8, 3.3, 3.8, 5.1, 7, 9.1, and 12.7 m. Subjects were provided with a simple task upon reaching the target: They were provided a pointer stick and instructed to walk to and touch the pointer to the target marker. This was intended to provide a context for the task, reflecting the fact that humans often walk to a particular destination to accomplish a task. Each distance condition was conducted a total of four times in two pairs of out-and-back trials, with the distances in random order. There were therefore a total of 400 trials, from 10 subjects walking 10 distances, each four times.

Walking speeds were measured from foot-mounted inertial measurement units (IMUs). These were used to compute the spatiotemporal trajectory of each foot in 3D, which was then processed to yield forward walking speed for the body per step. Each IMU (Opal sensors, APDM Inc, Portland, Oregon) was placed on the top of each foot, taped to the outside of the shoe. The recorded data of linear acceleration and angular velocity data were integrated using a previously described algorithm (*Rebula et al., 2013*) to yield foot trajectories. Briefly, the algorithm detects footfalls as instances in time when the foot is momentarily at rest on the ground, as defined by thresholds for acceleration and angular velocity. The footfall instance was defined as the mid-point of the below-threshold interval, and used to correct the integrated foot velocity (from gravity-corrected inertial accelerations) to zero, thus reducing IMU integration drift. The footfalls were also used to segment data into discrete strides, from which speed and length of each stride was calculated. (Subjects also wore another IMU on a waist belt, the data from which was used to demarcate the trials, but not for any further quantitative analysis.)

There were a few other analysis adjustments required to produce forward walking data. The absolute position and compass heading of the IMUs were unknown, yielding independent foot trajectories

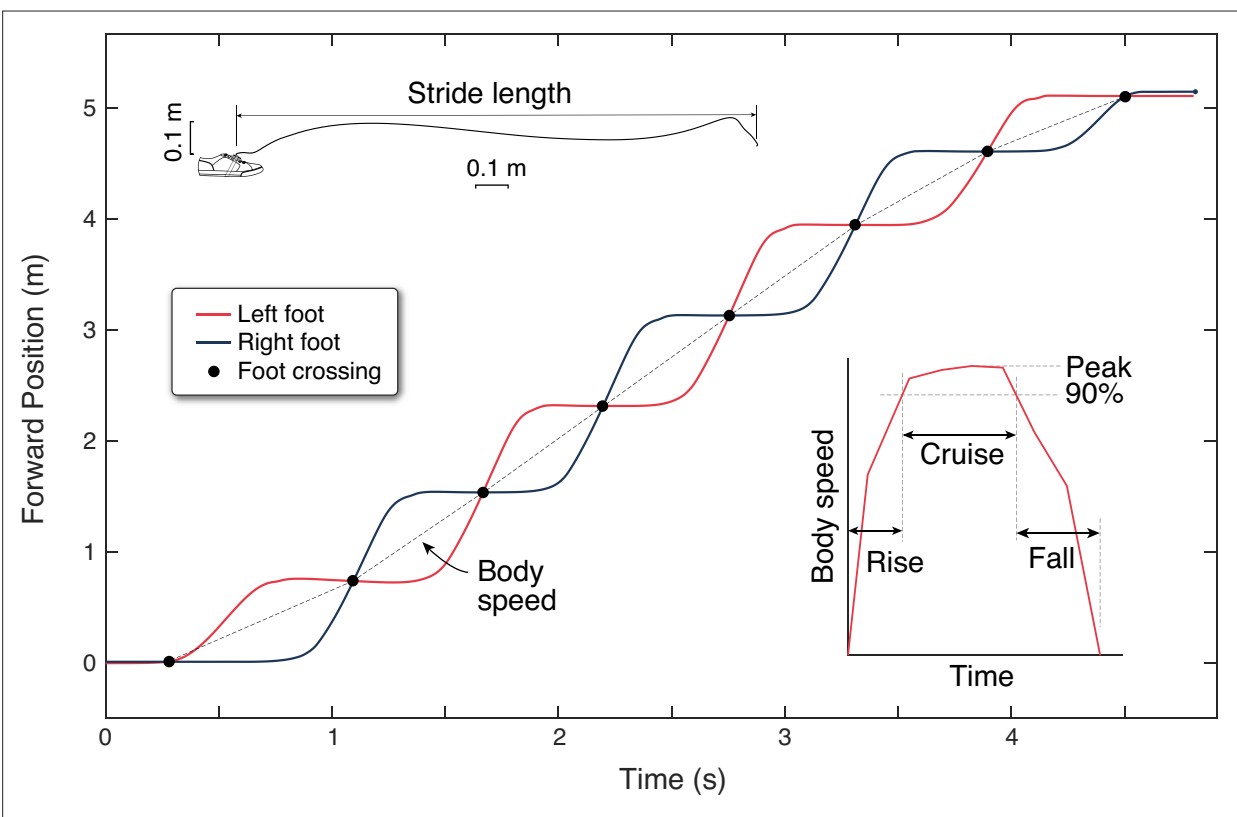

**Figure 8.** Experimental estimation of walking speed from inertial measurement units (IMUs). Forward position vs. time are shown for both feet (black and red lines) for a single walking bout of eight steps. Forward position is determined from foot trajectories, computed by integrating gravity-corrected inertial data (top inset). Each foot moves one stride length at a time, and the crossing points of the two feet define mid-stance instances that separate individual steps (black dots). Body speed is defined as the step length divided by step duration (slope of dotted line) for each step. Walking speed trajectories are plotted as discrete body speed vs. time. There are three durations defined within a walking bout (right inset): rise time, cruise time, and fall time. Rise and fall times are to accelerate from rest to 90% of peak speed and the converse. Cruise time is the time spent between 90% and peak speed.

with no relation to each other. However, the experimental conditions called for forward walking for a known distance, so we rotated each foot path to align them into a single forward direction. We also assumed that both feet travelled approximately the same distance for each walk, and translated and rescaled the start and end points to match each other, to yield a processed position-time graph of the two feet (see representative data in *Figure 8*). We also devised a definition for the starting and ending times for each trial based on IMU data. Humans initiate their gait by shifting their weight before moving the feet (*Mann et al., 1979*), so that the footfall threshold defined above may not detect the actual gait initiation. We therefore defined a rough approximation to gait initiation and termination, starting before and ending after threshold crossing, by an amount equal to half the average below-threshold time during walking. This adjustment may be incorrect compared to actual weight shift by several tens of milliseconds. The experiment is mainly concerned with speed profiles over time on the order of several seconds. The accuracy of the experiment can thus tolerate small errors in detecting gait initiation or termination.

Finally, the body's walking speed and length of each step were calculated as follows (*Figure 8*). The trajectory of each foot's strides were found to cross each other, approximating the time in mid-stance when one foot passes by the other. These points of intersection were used to define step length as the spatial distance between intersections, and step time as the temporal difference between inter-sections. These were used to determine the average speed for each step ('body speed'), defined as the step length divided by step time ending at each intersection. This assumes that the body moves as much as the feet between mid-stance instances. These discrete, step-averaged data were used to produce trajectories of body speed for each bout (*Figure 8*, inset), without regard to continuous-time undulations in velocity for the body center of mass. For comparison with these data, similar discrete body speeds and times were computed from model predictions.

We used these data to test the model predictions. We examined how human speed profiles varied with bout distance, and exhibited more rounded peaks for shorter bouts and flatter ones for longer ones. We tested for self-similarity by scaling the profiles by speed and time and performing statistical tests regarding peak speeds and walking durations. We tested whether a saturating exponential describes the increase in peak speed with bout distance ($R^2$; test 95% confidence interval of parameters not including zero). Expecting a self-similar shape for the peak speed vs. distance relationship, we scaled the curves by peak speed and tested for a single exponential. We tested self-similarity in terms of a reduction of variability in peak speed (standard deviation across subjects) for each condition, comparing non-normalized to normalized peak speeds (rescaled to mean overall peak speed) with paired t-test. We examined the walking durations as a function of bout distance, and also tested self-similarity by significant reduction in standard deviations across subjects, comparing non-normalized to normalized data (rescaled to mean longest duration) with paired t-test. We also described walking durations in terms of rise and fall times (between 10% and 90% of peak speed).

Prior to the experiment, subjects provided informed consent as approved by the University of Calgary Conjoint Health Research Ethics Board (REB21-1497). Pre-established exclusion criteria included significant health or other conditions that preclude ability to walk on uneven terrain or moderate hiking trails; no prospective participants were excluded. Subjects were recruited from the community surrounding the University of Calgary; the city has a moderately affluent population of about 1.4 M, with a developed Western culture. The experiment was performed once.

## Effect of valuation of time on steady walking speed

We performed an additional analysis to consider how the hypothesized energetic value of time may affect human steady walking speeds (*Figure 6*). This requires a valuation of time in terms of human metabolic energy rather than the model's mechanical work, and a consideration of longer walking bouts where steady walking dominates. To empirically quantify human cost as a function of speed, we fitted the model's steady mechanical work rate to human net metabolic power reported by *Elftman, 1966*, with a resting power adjusted to agree with the optimal steady speed of 1.25 ms$^{-1}$ reported by *Ralston, 1958*. The model was of the form

$$\dot{E}(v) = a \left( \frac{v + b}{\sqrt{gL}} \right)^n + d \tag{15}$$

where $a$, $b$ and $d$ are empirical coefficients, and $n$ is a model constraint. The exponent $n$ is not critical, and values ranging 2–4 are sufficient to describe the increase. However, we used a value of $n = 3.42$ as predicted by the simple model for human-like walking (**Kuo, 2002**). For metabolic power in, the empirical coefficients are $a = 4.90 \, \text{Wkg}^{-1}$, $b = 1.16 \, \text{ms}^{-1}$, and $d = 1.56 \, \text{Wkg}^{-1}$ ($R^2 = 0.99$). The y-intercept may be regarded as a resting rate, at $1.73 \, \text{Wkg}^{-1}$ (**Figure 6A**). The resulting cost is therefore proportional to the model's mechanical work, while matching well with human metabolic power and optimal steady speed data. The curve may be expressed as cost of transport by dividing power by speed, $\dot{E}/v$.

We then used our own walking data to estimate the human valuation of time. We used the peak walking speeds from the longest walking bout as indicator of steady speed. These were compared to the steady speed predicted by the metabolic cost curve with an added variable, the metabolic valuation of time $C_T$. The Energy-Time curve was converted to cost of transport, and then minimized to yield optimal speed. This is equivalent to taking the limit of the Energy-Time objective as function of increasing distance, thus making the costs of starting and ending a walking bout small. The result predicts that steady speed will increase approximately with the cube root of $C_T$ (**Figure 6B**). This curve was thus used to estimate $C_T$ for experimentally observed range of steady speeds. It was also used to estimate the effective cost of transport, including the valuation of time, as a function of speed (**Figure 6C**). This cost of transport may further be regarded as the sum of separate costs for Walking and Time (**Figure 6C**), where Walking refers to the cost of transport due to push-off work alone, and Time refers to the cost of transport due to the $C_T$ term alone.

## Additional information

### Funding

| Funder | Grant reference number | Author |
| --- | --- | --- |
| Natural Sciences and Engineering Research Council of Canada | CRC Chair | Arthur D Kuo |
| Université Lille 1 - Sciences et Technologies | Tier 1 | Arthur D Kuo |
| Natural Sciences and Engineering Research Council of Canada | Discovery award | Arthur D Kuo |
| Dr. Benno Nigg Research Chair | | Arthur D Kuo |

The funders had no role in study design, data collection and interpretation, or the decision to submit the work for publication.

### Author contributions

Rebecca Elizabeth Carlisle, Conceptualization, Data curation, Formal analysis, Investigation, Methodology, Software, Writing – original draft, Writing – review and editing; Arthur D Kuo, Conceptualization, Formal analysis, Funding acquisition, Methodology, Project administration, Resources, Software, Supervision, Validation, Visualization, Writing – original draft, Writing – review and editing

### Author ORCIDs

Arthur D Kuo http://orcid.org/0000-0001-5233-9709

### Decision letter and Author response

Decision letter https://doi.org/10.7554/eLife.81939.sa1
Author response https://doi.org/10.7554/eLife.81939.sa2

## Additional files

### Supplementary files
• MDAR checklist

### Data availability
Data (*Carlisle and Kuo, 2023*, https://github.com/kuo-lab/short_walk_experiment) and code (https://github.com/kuo-lab/simplelocomotionmodel, copy archived at *Kuo, 2023*) for this study are in publicly-accessible archives.

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
