## [Editor Report]

This valuable study presents a new optimal control cost framework to predict features of walking bouts, adding a cost function term proportional to the duration of the walking bout in addition to the conventional energetic expenditure term. The authors show solid evidence that predicted optimal trajectories from simulations qualitatively match walking data from human subjects. This study provides a foundation for future studies testing the validity of the energy-time hypothesis for human gait speed selection across different walking bouts in able-bodied and patient populations.

---

## [Decision Letter]

**Decision letter after peer review:**

Thank you for submitting your article "Humans dynamically optimize walking speed to save energy and time" for consideration by *eLife*. Your article has been reviewed by 2 peer reviewers, and the evaluation has been overseen by a Reviewing Editor and Joshua Gold as the Senior Editor. The following individual involved in the review of your submission has agreed to reveal their identity: Sam Burden (Reviewer #1)

Essential revisions:

1) The title asserts that "humans dynamically optimize walking speed to save energy and time". Directly substantiating this claim would require independently manipulating the (purported) energy and time cost of walking for human subjects, but these manipulations are not undertaken in the present study. What the Results actually report are two findings: a. (simulation) minimizing a linear combination of energy and time in an optimal control problem involving an inverted-pendulum model of walking bouts that (i) start and stop at rest and (ii) walk at constant speed yields a gently-rounded speed-vs-time profile (Figure 2A); b. (experiment) human subject walking bouts that started and stopped at rest had self-similar speed-vs-time profiles at several bout lengths after normalizing by the average duration and peak speed of each subject's bouts (Figure 4B). If the paper established a strong connection between (a.) and (b.), e.g. if speed-vs-time trajectories from the simulation predicted experimental results significantly better than other plausible models (such as the 'steady min-COT' and 'steady accel' models whose trajectories are shown in Figure 2A), this finding could be regarded as providing indirect evidence in support of the claim in the paper's Title. Accordingly, it would be more accurate to assert 'brief human walking bouts look like trajectories of an inverted-pendulum model that minimize a linear combination of energy and time' (although this is obviously too wordy). But unfortunately, the connection between (a.) and (b.) is only discussed qualitatively, and the other plausible models introduced in the Results are not revisited in the Discussion. The representative 'steady min-COT' trace in Figure 2A may be a real contender with the 'Energy-Time' trace for explaining the experimental results in Figure 4, but this candidate is rejected at the end of the third-to-last paragraph in the "Model Predictions" subsection of Results based on the vague rationale that is never revisited. Re-examining and discussing these points will be important in a subsequent revision.

2) An additional limitation of the approach not discussed in the manuscript is that a fixed step length was prescribed in the simulations. The "Optimal control formulation" subsection in "Methods" summarizes the results of a sensitivity analysis conducted by varying the fixed step length, but all results reported here impose a constant-step-length constraint on the optimal control problem. Although this is a reasonable modeling simplification for steady-state walking, it is less well-motivated for the walking bouts considered here that start and stop at rest. For instance, the representative trial from a human subject in Figure 8 clearly shows initiation and termination steps that differ in length from the intermediate steps (visually discernable via the slope of the dashed line interpolating the black dots). Presumably different trajectories would be produced by the model if the constant-step-length constraint were removed. It is unclear whether this change would significantly alter predictions from either the 'Energy-Time' or 'steady min-COT' model candidates. While the reviewers agreed that this change would entail substantial work that may be out of the scope of the present paper, they did agree that it is important to at least discuss this limitation.

3) Related to the above, the reviewer thought that the last two sentences in the abstract are too strong and should be rephrased to be more accurate and defensible.

4) Please explain the rationale for the competing objective functions (constant acceleration, in particular). The min-CoT objective appears supported by literature, but it was clearly less plausible for transient gait than the energy-time objective. The constant acceleration and min-CoT objectives did not seem like terribly challenging competing hypotheses to refute. Perhaps the authors can test or comment on objectives that produce realistic families of speed profiles to the energy-time hypothesis.

5) The rationale for defining rise and fall times based on 90% of the peak speed is unclear. The experimental acceleration profiles appear slightly steeper than the deceleration profiles, up to ~50% of the max speed. Consequently, the interpretation of rise and fall times may depend on this threshold. Can the authors justify their threshold or report the sensitivity of their conclusions to alternative thresholds?

6) As the authors have a strong history of using simple models to study the energetics of gait, a (possibly speculative) discussion on how individual differences in specific features of locomotion (stabilization, leg swing, etc…) may have on the estimated valuations of time may improve readers' evaluation of the immediate generalizability of the paper's conclusions.

7) In accordance with *eLife* policies, the authors need to release their data and simulation code, as well as provide details about their experimental apparatus.

8) The manuscript would benefit from careful proofreading prior to publication – the reviewers caught a number of typographical and grammatical errors.

*Reviewer #1 (Recommendations for the authors):*

The manuscript would benefit from careful proofreading prior to publication – I caught a number of typographical and grammatical errors.

Title

As discussed in my public review, I think the claim asserted in the title is too strong and should be rephrased to be more accurate/defensible.

Abstract

I think the last two sentences in the abstract are too strong and should be rephrased to be more accurate/defensible.

Introduction

– I think it is important to clarify that the results summarized in the opening sentences pertain to laboratory studies. Particularly given that a key motivator for the present study is real-world relevance, it would be valuable to clarify what is known from the lab vs in the wild.

– The (Hoyt and Taylor, 1981) cite appears to only report data on horses (not “other animals”) – please confirm/clarify.

Results

Model predictions

– It was unclear to me on my first couple of readthroughs that there is a constant step-length constraint enforced in the optimization problem – this constraint should be clearly specified in the display equation that contains the plain-language formulation of the optimal control problem and/or the adjacent text.

– Perhaps refer to the agent here as a “walker” rather than “human”, both for clarity and so as to not anthropomorphize the model.

– The concluding paragraph of this subsection contains a lot of weasel words and should be revised. https://en.wikipedia.org/wiki/Weasel_word

Figure 2

– Am I understanding correctly that the Energy-Time trace has a smaller integral in (B) than the other two traces? If so, I have two suggestions:

1. Include a subfigure/overlay showing the integral of Work for the three traces in (B) to emphasize this point;

2. In the sentence “Energy-time hypothesis predicts the least total work, whereas steady min-COT and steady acceleration require more overall work.”, the two clauses are redundant, and I think Time should be capitalized for consistency with the rest of the document.

Figure 4

– Plots with many overlapping opaque lines are difficult to read and slow to render. I think the ‘One subject, all trials’ traces are fine as-is, but I suggest instead one of the following revisions to the ‘All subjects, all trials’ traces:

1. Adjust trace opacity (rather than lightness) so that overlaps visually convey density – this makes the figure easier to read, but further degrades rendering speed;

2. Replace traces with filled region(s) / trace(s) illustrating percentiles of data spread, e.g. interquartile [25%,75%] (and/or all-data [0%,100%]) fill with median (50%) (and/or quartile (25%,75%)) trace(s).

Figure 5

– Are the ‘Individuals’ traces shown pre- or post-normalization?

– The contrast between ‘raw’ and ‘normalized’ whiskers is difficult to discern.

– The visualization of Fall/Cruise/Rise times in (B) is fantastic!

Figure 6

– I found the phrase “virtual cost” confusing – isn’t the point of this paper to show that energy and time costs are on equal footing? I think the sentence is clearer if “is a virtual cost and” is removed.

Discussion

– It wasn’t clear to me why “time spent at destination is far more valuable than resting” in a big city – it would be helpful if the authors could clarify this claim with reference to the underlying literature.

– The idea in the penultimate paragraph to further test the Energy-Time hypothesis by manipulating the energetic cost of walking is clever! I hope the authors pursue this

– I suggest replacing "should" (which connotes authority) with "presumably" or "may" (to convey that this is a hypothesis, not an obvious truth).

Conclusion

– The phrase "we found it more general" is vague and (in my reading) not well-substantiated by the results. I suggest revising/rephrasing.

– It might be fun to cite the (purported) origin of the "time is money" aphorism.

https://en.wikipedia.org/wiki/Time_is_money_(aphorism)

Methods

– I appreciated the care that clearly went into the simulation and experimental study designs and the clear and concise writing in this section. In particular, I appreciated the discussion of tolerable errors in the timing of gait initiation and termination.

– It may be better to refer t“ "step-averaged body speed" – I was confused for a long time during my first read by the coarse and irregular time quantization in the results figures. Only once I read and internalized the simulation and experiment methods did I understand (i) the cleverness of the study design and (ii) how to interpret the figures.

– I was absolutely delighted to learn about JuMP! Thank you for teaching me – I commend your use of open-source tools and hope you’ll make your source code available to the community.

*Reviewer #2 (Recommendations for the authors):*

Please explain the rationale for the competing objective functions (constant acceleration, in particular). The min-CoT objective appears supported by literature, but it was clearly less plausible for transient gait than the energy-time objective. The constant acceleration and min-CoT objectives did not seem like terribly challenging competing hypotheses to refute. Perhaps the authors can test or comment on objectives that produce realistic families of speed profiles to the energy-time hypothesis.

The rationale for defining rise and fall times based on 90% of the peak speed is unclear. The experimental acceleration profiles appear slightly steeper than the deceleration profiles, up to ~50% of the max speed. Consequently, the interpretation of rise and fall times may depend on this threshold. Can the authors' test justify their threshold or report the sensitivity of their conclusions to alternative thresholds?

As the authors have a strong history of using simple models to study the energetics of gait, a (possibly speculative) discussion on how individual differences in specific features of locomotion (stabilization, leg swing, etc…) may have on the estimated valuations of time may improve readers' evaluation of the immediate generalizability of the paper's conclusions.

It would be interesting to see the following results, probably in a supplemental:

1. The individual values of cT for human participants and the corresponding preferred speeds.

2. Subject-specific trajectories from Figure 4. Do any participants exhibit unexpected behavior?

---

## [Author Response]

Essential revisions:1) The title asserts that "humans dynamically optimize walking speed to save energy and time". Directly substantiating this claim would require independently manipulating the (purported) energy and time cost of walking for human subjects, but these manipulations are not undertaken in the present study. What the Results actually report are two findings: a. (simulation) minimizing a linear combination of energy and time in an optimal control problem involving an inverted-pendulum model of walking bouts that (i) start and stop at rest and (ii) walk at constant speed yields a gently-rounded speed-vs-time profile (Figure 2A); b. (experiment) human subject walking bouts that started and stopped at rest had self-similar speed-vs-time profiles at several bout lengths after normalizing by the average duration and peak speed of each subject's bouts (Figure 4B). If the paper established a strong connection between (a.) and (b.), e.g. if speed-vs-time trajectories from the simulation predicted experimental results significantly better than other plausible models (such as the 'steady min-C’T' an‘ 'steady accel' models whose trajectories are shown in Figure 2A), this finding could be regarded as providing indirect evidence in support of the claim in the paper's Title. Accordingly, it would be more accurate to assert 'brief human walking bouts look like trajectories of an inverted-pendulum model that minimize a linear combination of energy and time' (although this is obviously too wordy). But unfortunately, the connection between (a.) and (b.) is only discussed qualitatively, and the other plausible models introduced in the Results are not revisited in the Discussion. The representative 'steady min-C’T' trace in Figure 2A may be a real contender with the 'Energy-Time' trace for explaining the experimental results in Figure 4, but this candidate is rejected at the end of the third-to-last paragraph in the "Model Predictions" subsection of Results based on the vague rationale that is never revisited. Re-examining and discussing these points will be important in a subsequent revision.

We agree with the comments about title, and have changed the title to “Optimization of energy and time determines dynamic speeds for human walking”. We believe this is accurate and consistent with the reviewer’s concerns.

We have also clarified why we reject the min-COT hypothesis. In Experimental Results, “In contrast to the min-COT hypothesis, the human peak speeds increased with distance, many well below the min-COT speed of about 1.25 m/s.” (The min-COT hypothesis always predicts only one speed.) And in Discussion, “Shorter bouts… exhibited a relatively brief peak slower than the typical min-COT speed… All such bouts also spent substantial time and energy in acceleration and deceleration…This contrasts with the min-COT hypothesis, which predicts only a single steady speed, and cannot explain how to start or end a bout.”

2) An additional limitation of the approach not discussed in the manuscript is that a fixed step length was prescribed in the simulations. Th“ "Optimal control formulation" subsection in "Methods" summarizes the results of a sensitivity analysis conducted by varying the fixed step length, but all results reported here impose a constant-step-length constraint on the optimal control problem. Although this is a reasonable modeling simplification for steady-state walking, it is less well-motivated for the walking bouts considered here that start and stop at rest. For instance, the representative trial from a human subject in Figure 8 clearly shows initiation and termination steps that differ in length from the intermediate steps (visually discernable via the slope of the dashed line interpolating the black dots). Presumably different trajectories would be produced by the model if the constant-step-length constraint were removed. It is unclear whether this change would significantly alter predictions from either the 'Energy-Time' o‘ 'steady min-C’T' model candidates. While the reviewers agreed that this change would entail substantial work that may be out of the scope of the present paper, they did agree that it is important to at least discuss this limitation.

We were not sufficiently we had also included step lengths increasing according to the preferred human step length vs. speed relationship. We have made several changes to clarify, in Model Predictions, Figure 3, optimal formulation in Methods, and Discussion. The specific changes are listed below (emphasis is used here only, and is not in actual text).

a. We have added “at human-like step length” to the plain-language formulation.

b. In Model Predictions we have added, “Step length was nominally kept fixed, and then varied in parameter sensitivity studies below.”

c. In Figure 3 caption, “In the main plot, multiple predictions for different time valuations and *step lengths* are scaled and superimposed on each other… Three different step lengths including shorter and longer steps than nominal, and human preferred step length relationship (dashed lines); main plot also includes nominal step length.”

d. In the parameter sensitivity paragraph, “We considered step lengths s fixed at nominal (0.68 m), at slightly shorter and longer lengths (0.59 m and 0.78 m), and increasing with speed according to the human preferred step length relationship (see Methods for details).

e. In Methods, we moved the description of step length from the end of the sub-section to earlier, and added a passage to clarify our hypothesis: “Step lengths were examined with a nominal fixed step length of 0.68 m, and sensitivity analyses performed with fixed lengths of 0.59 m and 0.78 m, *and varying lengths following the human preferred step length relationship* s=v^0.42 (Grieve, 1968). We have previously proposed that step-to-step transitions account for most of the metabolic cost of human walking (Donelan et al., 2002), and that forced swing leg motions *to modulate step length also contribute a non-negligible, but* lesser cost (Kuo, 2001) that is neglected here. This is not to dismiss this and other costs of locomotion, but merely to hypothesize that step-to-step transitions will still dominate in transient walking bouts. *The failure to include other costs, if sufficiently critical, should cause the model to make poor predictions.*”

f. In Discussion, we have clarified that the mention of unmodeled forced leg motion is related to “step length modulation”.

3) Related to the above, the reviewer thought that the last two sentences in the abstract are too strong and should be rephrased to be more accurate and defensible.

We have reworded as follows (see emphasis):

“Individual-dependent vigor *may be* characterized by the energy one is willing to spend to save a unit of time, which explains why some may walk faster than others, but everyone *may have* similar-shaped trajectories due to similar walking dynamics. Tradeoffs between energy and time costs *can* predict transient, steady, and vigor-related aspects of walking.”

4) Please explain the rationale for the competing objective functions (constant acceleration, in particular). The min-CoT objective appears supported by literature, but it was clearly less plausible for transient gait than the energy-time objective. The constant acceleration and min-CoT objectives did not seem like terribly challenging competing hypotheses to refute. Perhaps the authors can test or comment on objectives that produce realistic families of speed profiles to the energy-time hypothesis.

We have added more description of the steady acceleration hypothesis to Methods:

“The second alternative was a steady acceleration hypothesis, which contrasts with min-C’T's immediate high acceleration. Here the acceleration is made as gradual as possible, albeit at the expense of a higher peak speed needed to travel the same distance and duration. Both of these alternatives help illustrate how different speed trajectories requires differing amounts of mechanical work, to be compared against the work produced by the Energy-Time hypothesis.”

We have added a Discussion paragraph about alternative models:

“It is instructive to consider what other models might explain or predict our experimental results. We did not explore more complex models here, but would expect similar predictions to result from any model based on pendulum-like walking and step-to-step transitions. This includes those cited in the previous paragraph, as well as a family of such models including three-dimensional motion…knees and bi-articular actuators…, and plantarflexing ankles… It is also possible that more complex, musculoskeletal models also perform substantial work and expend energy for step-to-step transitions, and might therefore agree with the present model. But here, relatively simple principles account for a fairly wide array of predictions (Figure 3), which are unlikely to result from happenstance. We are also unaware of any current hypothesis that could plausibly substitute for the present one. We therefore doubt if alternative models not based on pendulum-like principles could predict or reproduce these results, except with numerous fitted parameters.”

5) The rationale for defining rise and fall times based on 90% of the peak speed is unclear. The experimental acceleration profiles appear slightly steeper than the deceleration profiles, up to ~50% of the max speed. Consequently, the interpretation of rise and fall times may depend on this threshold. Can the authors justify their threshold or report the sensitivity of their conclusions to alternative thresholds?

There is nothing fundamental about the definition of rise/fall times. To clarify that these measures are only used descriptively, we have amended the Results to say “This was *described* by rise time…” which recapitulate the Methods “We also *described* walking durations in terms of rise and fall times” (emphasis is not in the actual text). The definition here is drawn from the convention in control systems (10% and 90%, https://en.wikipedia.org/wiki/Rise_time), but other definitions could also be appropriate.

We made no strong claims based on these measures, which were defined mainly to provide some visualization (Figure 5B). The only statements were (in Results), “As a fraction of each bout's duration, the rise and fall times *appeared to* take up a greater proportion for shorter bouts, and only a very small proportion was spent at steady speed.

Conversely, cruise time took up a greater proportion of the time for longer bouts.” The “appeared to” was intentional, to indicate a qualitative observation based on non-fundamental rise/fall times. The second sentence is a continuation of the first, and so we did not bother add another qualifier there. We feel this is a defensible statement, because all trials from all subjects are shown in Figure 4, and we do not think the observation is highly sensitive to the definition of rise/fall time.

6) As the authors have a strong history of using simple models to study the energetics of gait, a (possibly speculative) discussion on how individual differences in specific features of locomotion (stabilization, leg swing, etc…) may have on the estimated valuations of time may improve readers' evaluation of the immediate generalizability of the paper's conclusions.

We have expanded the speculative Discussion of energetics and stability:

“The simple walking model also only includes a crude representation of step-to-step transitions, which we have crudely estimated to account for as much as 70% of net metabolic cost in steady walking (Donelan et al., 2002). We did not include other factors such as forced leg motion and step length modulation (up to say 33% of cost, Doke et al. 2005) that likely also affect energetic cost, and could therefore be used to test the valuation of time. Nor did we include factors such stability (Bauby and Kuo, 2000; Donelan et al., 2004, Rebula et al., 2017) and three-dimensional motion Donelan et al., 20010, despite being part of our previous models, because we believe them to contribute little to the present task. In fact, because the optimal control model successfully completes the walking task, its feedforward motion attains implicit stability, which could reduce (but not eliminate) the need for feedback stabilization (Darici and Kuo, 2022 a, b).”

7) In accordance with eLife policies, the authors need to release their data and simulation code, as well as provide details about their experimental apparatus.

Yes, we intend to share both data and optimization code if manuscript is accepted, as indicated in Data Availability statement. This information is under preparation.

8) The manuscript would benefit from careful proofreading prior to publication -- the reviewers caught a number of typographical and grammatical errors.

We apologize for the errors and have made an effort to fix them.

Reviewer #1 (Recommendations for the authors):

Almost all of the following has been addressed above in Editor comments. Here we respond to a few comments with additional information.

Conclusion– It might be fun to cite the (purported) origin of the "time is money" aphorism.https://en.wikipedia.org/wiki/Time_is_money_(aphorism)

The origin of the aphorism is interesting, but we prefer not to include it.

Methods– I was absolutely delighted to learn about JuMP! Thank you for teaching me – I commend your use of open-source tools and hope you'll make your source code available to the community.

We do intend to share the code and the data. The entire toolchain for the optimization is open source.

Reviewer #2 (Recommendations for the authors):

All of this has been addressed above in Essential comments.